# Transition Mutations in the hTERT Promoter Are Unrelated to Potential i-motif Formation in the C-Rich Strand

**DOI:** 10.3390/biom13091308

**Published:** 2023-08-25

**Authors:** James W. Conrad, Mark L. Sowers, Dianne Y. Yap, Ellie Cherryhomes, B. Montgomery Pettitt, Kamil Khanipov, Lawrence C. Sowers

**Affiliations:** 1Department of Pharmacology and Toxicology, The University of Texas Medical Branch, Galveston, TX 77555, USA; 2MD-PhD Combined Degree Program, The University of Texas Medical Branch, Galveston, TX 77555, USA; 3Department of Biochemistry and Molecular Biology, The University of Texas Medical Branch, Galveston, TX 77555, USA; 4Department of Internal Medicine, The University of Texas Medical Branch, Galveston, TX 77555, USA

**Keywords:** telomere, telomerase, cytosine deamination, transition mutation, quadruplex

## Abstract

Increased expression of the human telomere reverse transcriptase (hTERT) in tumors promotes tumor cell survival and diminishes the survival of patients. Cytosine-to-thymine (C-to-T) transition mutations (C250T or C228T) in the *hTERT* promoter create binding sites for transcription factors, which enhance transcription. The G-rich strand of the *hTERT* promoter can form G-quadruplex structures, whereas the C-rich strand can form an i-motif in which multiple cytosine residues are protonated. We considered the possibility that i-motif formation might promote cytosine deamination to uracil and C-to-T mutations. We computationally probed the accessibility of cytosine residues in an i-motif to attack by water. We experimentally examined regions of the C-rich strand to form i-motifs using pH-dependent UV and CD spectra. We then incubated the C-rich strand with and without the G-rich complementary strand DNA under various conditions, followed by deep sequencing. Surprisingly, deamination rates did not vary substantially across the 46 cytosines examined, and the two mutation hotspots were not deamination hotspots. The appearance of mutational hotspots in tumors is more likely the result of the selection of sequences with increased promoter binding affinity and *hTERT* expression.

## 1. Introduction

The human telomere reverse transcriptase (hTERT) replicates the ends of human telomeres, reversing telomere attrition and replicative senescence that would occur in the absence of hTERT activity. The expression of *hTERT* promotes cell survival, but its overexpression in many human tumors renders them immortal [1]. In the highly lethal and incurable brain tumor glioblastoma, Barthel et al. (2017) found that 89% of tumors had *TERT* promoter mutations, primarily C-to-T transitions, and were found at two predominant sequence locations [2]. Recurrent mutations at C250T and C228T in glioblastoma are positively correlated with increasing age at diagnosis [3]. Simon et al. (2015) found that C-to-T transition mutations were associated with overall poor survival in glioblastoma patients with incomplete resection and those who had not been treated with temozolomide [4]. Combined analysis of *TERT* mutations within the context of the status of other genes, including *EGFR* and *IDH*, can further refine the prognostic classification of glioblastoma [5].

Activation of *hTERT* expression results from C-to-T transition mutations at either of two specific sites in the CG-rich hTERT promoter (Figure 1). The presence of the resulting TT dinucleotide at specific sequence positions substantially increases the binding affinity of two transcription factor families, the E-twenty-six (ETS) and Specificity Protein/Kruppel-Like Factor groups, with subsequent increases in *hTERT* expression [6,7,8,9]. Currently, mechanisms that would promote C-to-T transition mutations at these sites are unknown.

Mutational hotspots in the *hTERT* promoter are within a C- and G-rich region that can form alternative structures. The G-rich strand has been extensively studied in the absence of the complementary C-rich strand, and it is known to form a G-quadruplex structure under physiological conditions [10]. The G-rich strand has 12 runs of three or more G’s that can form multiple quadruplex structures that mask Sp1 binding sites and likely contribute to diminished *hTERT* transcription. In the tandem G-quadruplex structure proposed by Palumbo et al. (2009), the G’s, which would be paired with C250 and C228 in a normal duplex, would be involved in the tandem quadruplex and unavailable to form Watson–Crick base pairs with the hotspot C’s [10].

The complementary C-rich strands of the *BCL2* and *KRAS* promoters can form an i-motif structure below physiological pH, where some cytosine residues can be protonated [11]. Equilibrium among multiple structures when the duplex is present is supported by experimental data [12]. Cui et al. (2016) suggest that the formation of a G-quadruplex by the G-rich strand and i-motif formation by the C-rich strand from a complementary duplex are mutually exclusive due to steric hindrance [13]. The potential contribution of alternative, non-duplex DNA structures to C-to-T mutations has not been considered previously.

Cytosine bases in DNA are vulnerable to hydrolytic deamination with the formation of uracil [14]. The C250T and C228T transition mutations that activate *hTERT* likely arise from cytosine deamination. If unrepaired, uracil arising from cytosine deamination codes as thymine in the next round of DNA replication, resulting in a C to T transition mutation. Cytosine-to-thymine transition mutations are frequently found in human tumors [15]. Hotspots for transition mutations are observed at CpG dinucleotides, where the C is usually methylated, forming 5-methylcytosine (5 mC) [16]. Unlike uracil (U) generated from C deamination, the T generated from 5 mC deamination is poorly repaired by human glycosylases [17], accounting in part for the hotspot frequencies at CpG dinucleotides [18]. However, the C250 and C228 mutation hotspots in the *hTERT* promoter are not at CpG sites, and therefore 5 mC deamination and repair would not explain why these sites are mutational hotspots. While uracil repair within the hTERT promoter has not been investigated, it is known that the completion of DNA repair by the base excision-repair pathway can be inhibited when uracil bases are located within DNA sequences like telomeres that can form non-duplex structures [19,20,21]. Inefficient BER within the *hTERT* promoter could contribute to increased mutation frequencies.

The mechanism of cytosine deamination involves a water or hydroxide attack on cytosine at the C4 position, followed by displacement of the four-amino group. Factors considered important in determining the rate of deamination include cytosine protonation, water accessibility and temperature [22,23,24,25,26]. Previously, the rates of deamination of cytosine and analogs have been reported. Lyndahl and Nyberg (1974) measured the rate constants for cytosine deamination in single-stranded DNA, poly(dC) and dCMP to be 2 × 10^−7^ s^−1^ at pH 7.4 and 95 °C [14]. Cytosine deamination in duplex DNA was reduced by a factor of 100 and attributed to protection from water in the duplex structure. Surprisingly, the rate of cytosine deamination in poly(dG)-poly(dC) was 75% that observed for single-stranded DNA. The rate of cytosine deamination at physiological pH and temperature is too slow to measure using standard chemical assays. Ehrlich et al. (1986) measured the deamination of cytosine in single-stranded DNA at higher temperatures and extrapolated the rate constant at 37 °C to be 2 × 10^−10^ s^−1^ [27]. Frederico et al. (1993) developed a genetic assay and measured cytosine deamination rate constants in single-stranded and duplex DNA at pH 7.4, 37 °C, to be 1 × 10^−10^ and 7 × 10^−13^ s^−1^ [28]. Shen et al. (1994) similarly used a genetic reversion assay to measure the rate constant for cytosine deamination in duplex DNA to be 2.6 × 10^−13^ s^−1^ [29].

Cytosine bases in the C-rich strand of the *hTERT* promoter could potentially be found in a Watson–Crick base pair in duplex DNA, single-stranded DNA within loops, or in a hemi-protonated base pair of an i-motif. Cytosine deamination in a hemi-protonated i-motif has not been investigated previously. While protonation would be expected to increase the rate of deamination substantially [24], if the cytosine C4 position is protected from solvent in the i-motif, the deamination rate might be less than in single-stranded DNA despite protonation.

In the study presented here, we were interested in determining if the possible environments for cytosine bases within the C-rich strand of the *hTERT* promoter might be related to deamination rates and, more specifically, if such structures would contribute to the hotspot frequency of mutations observed at C250 and C228. In the first part of this study, we used computational methods to explore the solvent accessibility of the C4 position of cytosines located in a model i-motif configuration. We then used spectroscopic studies on a series of oligonucleotides derived from the *hTERT* promoter to determine the propensity of defined regions to form an i-motif structure. Finally, we incubated a 123-base oligonucleotide (Figure 2) containing the 68-base CG-rich region of the *hTERT* promoter under defined conditions of pH and temperature, with and without the complementary G-rich strand, and measured deamination simultaneously at all 46 cytosine positions using a next-generation sequencing method developed for this study.

## 2. Material and Methods

### 2.1. Computational Studies

3D models of an i-motif (PDB: 1ELN) [30] or duplex DNA (PDB: 1CGC) [31] were obtained from the RSCB Protein Data Bank (PDB). Models were imported into Visual Molecular Dynamics (VMD v1.94a53) [32], and solvent accessible surface area (SASA) was calculated for entire molecules, individual bases, individual carbon 4, 5, or 6 atoms within a cytosine, or other combinations using a 1.4 Å radius to represent a water molecule.

### 2.2. Oligonucleotide Synthesis and Purification

The full-length *hTERT* core promoter oligonucleotides and primers were purchased from Integrated DNA Technologies (Appendix A). Duplexes were prepared by annealing full-length oligonucleotides in Britton–Robertson buffer at a 1:1 ratio and subsequently diluted for experiments. Annealing was performed by heating to 95 °C for 1 min and then slowly cooling to room temperature. Defined-sequence oligonucleotides were synthesized by solid-phase phosphoramidite methods (Appendix A). Sequences were verified by Maldi-MS (Appendix A). Oligonucleotide concentrations were determined using A_260_ extinction coefficients obtained from Oligo Calc (Northwestern.edu) and an Implen NanoPhotometer^TM^ (Implen, Munich, Germany).

### 2.3. Examination of Oligonucleotide Configurations in Aqueous Solution Using CD and UV

**Buffer:** Britton–Robertson buffer (BRB): 20 mM sodium borate, 20 mM sodium phosphate, 20 mM sodium acetate, 140 mM KCl, pH adjusted with concentrated HCl. The total volume was changed by less than 5% following titration.

**i-motif CD:** Partial-length *hTERT* oligonucleotide fragments (Appendix A) were used to determine the location and capacity for i-motif formation in the full-length sequence. Each oligonucleotide was diluted to 2 μM in BRB with a pH ranging from 4 to 8. Samples (500 μL) were equilibrated at room temperature for ≥30 min prior to taking measurements. The formation of an i-motif produces a characteristic peak near 285 nm [33,34]. Circular dichroism (CD) experiments were measured in a 100-μL cuvette (Starna, cat# 26.100LHS-Q-10/Z15) with a 1 cm path length and a Jasco J-815 CD spectrophotometer equipped with a temperature controller (CDF-426S) set to 20 °C. Spectra were acquired from 320 to 220 nm with a data pitch of 0.5 nm, standard sensitivity, D.I.T. 4 s, a bandwidth of 2 nm, and a scan speed of 100 nm/min. Samples were baseline corrected using a buffer-only control. Data were smoothed in Spectra Manager using the Savitsky–Golay filter with a window size of 20. Values were adjusted for the isosbestic point and then imported into Prism for analyses. The transition point, p*K_a_*, was then determined for each oligo.

**i-motif melt curve CD:** Partial and full-length *hTERT* oligonucleotides were diluted to 2 μM in BRB at pH 5, 6, and 7. Samples (500 μL) were equilibrated at room temperature for ≥30 min prior to taking measurements. Circular dichroism experiments were measured in a 100-μL cuvette (Starna, cat# 26.100LHS-Q-10/Z15) with a 1 cm path length and a Jasco J-815 CD spectrophotometer equipped with a temperature controller (CDF-426S) set to ramp from 20 to 90 °C. CD was measured at 285 ± 5 nm with a temperature ramp rate of 2 °C per minute. Samples were baseline corrected using a buffer-only control. The melting temperatures were determined in Graphpad PRISM v10.0.1 by taking the first derivative.

### 2.4. Oligonucleotide Incubation to Increase C-to-T Mutations

The single-stranded and duplex Seq-hTERT-C/G-rich oligonucleotides (Appendix A) were used in sequencing experiments. Incubations were conducted in 100 μL reactions containing 0.5 μM DNA (~20 or ~40 ng/μL for ssDNA and duplex, respectively) in BRB (pH 5, 6, or 7). The samples were incubated at 37, 45, or 95 °C for up to 11 days. Each incubation vial had 50 μL of mineral oil added to reduce evaporation. At each time point, two 10-μL aliquots of the sample were taken and stored at 4 °C. When the last time point was collected, a 10-μL aliquot from each time point was used for library preparation and sequencing. The duplex oligonucleotide was treated similarly.

### 2.5. Detection of Cytosine Deamination by Sequencing

**Library preparation for Next-Generation Sequencing:** The incubated single-stranded or duplex DNA samples were quantified by Qubit using a ssDNA (cat# Q10212, Thermo Fisher Scientific, Waltham, MA, USA) or 1× dsDNA High Sensitivity (cat# Q33233, Thermo Fisher Scientific, Waltham, MA, USA) Qubit Assay kit. The samples were diluted with 10 mM Tris-HCl pH 8.5 (cat# T1062, Teknova, Hollister, CA, USA) to 5 ng/μL in preparation for a two-step targeted amplicon library preparation method adapted from the 16S Metagenomics Sequencing Library Preparation protocol by Illumina (Part# 15044223 Rev. B, Illumina, San Diego, CA, USA). In the first PCR step, a 25 μL reaction mixture consisting of 12.5 ng of ssDNA or duplex DNA template, 0.3 μM each of custom forward and reverse primers (IDT, Coralville, IA, USA) that amplify the target 200 bp hTERT with indexing adapter binding site sequence, and 1× of KAPA Hifi HotStart Uracil + ReadyMix Kit (cat# KK2801, Roche, Basel, Switzerland) was prepared for each treatment. The target sequences were then amplified using a Thermo Scientific™ Arktik thermocycler (cat# TCA0002PROM, Thermo Fisher Scientific, Waltham, MA, USA) with the following PCR program: initial denaturation at 95 °C for 3 min; followed by 25 cycles of 95 °C for 20 s, 65 °C for 15 s, and 72 °C for 30 s; and a final extension at 72 °C for 5 min with a 4 °C hold. Primer dimers, PCR enzymes and buffer components were removed from the PCR products by using 1.8× AMPureXP beads (cat# A63381, Beckman Coulter, Brea, CA, USA) to 1× PCR reaction volume clean-up. The AMPureXP paramagnetic beads bind the 200-bp target DNA while contaminants are removed by washing the beads with 80% ethanol. The purified DNA was then eluted in 52.5 μL of 10 mM Tris-HCl pH 8.5 buffer.

For the second PCR step of the library preparation, a unique dual 8-bp index combination was assigned to each ssDNA or duplex DNA incubation, which allows the loading of multiple identifiable libraries in one sequencing run. For this step, 50 μL PCR reactions were prepared by combining 5 μL of the purified first step PCR product, 5 μL each of the assigned Index 1 and Index 2 adapter combinations from primer set B of the NexteraXT Index Kit v2 (cat# FC-131-2002, Illumina, San Diego, CA, USA), and 1× KAPA HiFi HotStart ReadyMix (cat# KK2602, Roche, Basel, Switzerland). The PCR reactions were placed in the thermocycler with a PCR program that consists of an initial denaturation at 95 °C for 3 min; 8 cycles of 95 °C for 30 s, 55 °C for 30 s, and 72 °C for 30 s; and a final extension at 72 °C for 5 min. The PCR products were purified using a 1.8:1 AMPureXP beads-to-DNA volume ratio, and the sequencing libraries were eluted in 27.5 μL of 10 mM Tris-HCl pH 8.5 buffer. After each AMPureXP bead cleanup, the size of the PCR product for each treatment was verified using the Agilent 2100 Bioanalyzer system (cat# G2939A, Agilent Technologies, Santa Clara, CA, USA) with the DNA 1000 Bioanalyzer Chip kit (cat# 5067-1504, Agilent Technologies, Santa Clara, CA, USA).

Concentrations of the resulting sequencing libraries were measured using the 1× dsDNA Qubit High Sensitivity Assay Kit (cat# Q33233, Thermo Fisher Scientific, Waltham, MA, USA). Each sequencing library was diluted to 4 nM with 10 mM Tris-HCl pH 8.5, and equimolar amounts of the sequencing libraries were pooled together. The pooled library was then denatured in 0.1 N NaOH at room temperature for 5 min. This was further diluted to 4 pM using chilled HT1 hybridization buffer (20015892, Illumina, San Diego, CA, USA). To increase nucleotide diversity in each sequencing cycle, a PhiX sequencing control library (cat# FC-110-3001, Illumina, San Diego, CA, USA) was denatured and diluted to 4 pM in the same way as the pooled library, and the PhiX control was combined with the sequencing library at 20% volume. Prior to sequencing, the final combined library was heat denatured at 96 °C for 2 min, then immediately placed on ice for at least 5 min. The library was then loaded onto a 300-cycle paired-end read MiSeq v2 reagent cartridge (cat# MS-102-2002, Illumina, San Diego, CA, USA) and sequenced on the MiSeq platform (cat# SY-410-1003, Illumina, San Diego, CA, USA).

**Processing sequencing reads:** bcl2fastq v1.8.4 software (Illumina, San Diego, CA, USA) was used on the Illumina BaseSpace to convert and demultiplex the bcl files to FASTQ. CLC Genomics Workbench 22.0.2 (Qiagen, Hilden, Germany) was used to analyze and process the sequencing data. FASTQ files were imported from the Illumina BaseSpace cloud. Trim Reads v2.6 was used to trim adapters from the reads and perform quality control. For the first paired reads, the read adapter “CAACTAC” was used to trim the 5′ end (allowing for no mismatches), and “GAG” was used to trim the 3′ end (allowing for no mismatches). For the second paired reads, the read adapter “GACAGCTC” was used to trim the 5′ end (allowing for no mismatches in the last three bases), and “GTAGTTG” was used to trim the 3′ end (allowing for no mismatches). Reads that did not contain the adapter sequences were discarded. Sequences containing any ambiguous nucleotides or lengths less than 50 bases were removed. Map Reads to Reference v1.8 were used to align the trimmed reads to the 123-nucleotide oligonucleotide sequence containing the hTERT core promoter (i.e., Seq-hTERT-C-rich (Appendix A)).

Global alignment was performed with a length and similarity fraction of 0.9 based on the known sequence, a default match score of 1, a mismatch score of 2, and an insertion/deletion cost of 3. Basic Variant Detection v2.3 was used to identify the frequency of C > T mutations. The following parameters were used to call a variant: (1) the minimum coverage per position was 10 reads, and (2) there was a minimum of 2 reads and a 0.00001% frequency for each mutation. Variant track data were exported into Excel files for further analysis. Paired-end sequencing data were initially QC-filtered and analyzed for the frequency of C > T mutations. However, a high frequency of insertions and deletions substantially reduced the number of usable reads [35,36] due to the repetitive nucleotides increasing the error [37,38]. Therefore, most data were analyzed by separating the paired-end read data into read 1 (forward) and read 2 (reverse). The frequency of C > T mutations at each nucleotide position within the 123-bp sequence was then plotted for each incubation condition. A comparison of average deletion frequencies was performed by analyzing data from read 1.

### 2.6. Statistical Analysis

The robust regression and outlier removal (ROUT) method was used to identify whether the transition mutations observed in the sequencing data were similar [39]. The background transition rates measured at pH 7 and 4 °C (ssDNA or dsDNA, as appropriate) were subtracted for each condition. Transition mutation rates for each cytosine position were analyzed using GraphPad Prism v10.0.1. A Q value of 1% was set for the ROUT analysis.

Mann–Whitney test [40] was used to compare the distributions of the transition mutation rates between the C and A/G/T positions across the hTERT promoter regions within each treatment condition.

Statistical comparison of the transition rates was performed using the Kruskal–Wallis test [41]. Background-subtracted transition mutation rates at each cytosine position were placed into groups according to their treatment condition. The non-parametric test, Kruskal–Wallis, was employed to compare the transition mutation rates across the different conditions. This test was chosen due to its ability to compare more than two independent groups without the assumption of a normal distribution. Following the Kruskal–Wallis test, Dunn’s multiple comparison test [42] was performed to identify which specific pairs of conditions had statistically significant differences in their mutation rates. This post-hoc test was essential to controlling the family-wise error rate and providing a pairwise comparison between the groups. Kruskal–Wallis and Dunn’s test results were visualized using box plots in GraphPad Prism v10.0.1.

## 3. Results and Discussion

### 3.1. Oligonucleotide Computational Studies

Mutations that activate the *hTERT* promoter most likely arise from the deamination of specific cytosine bases in the C-rich strand resulting from spontaneous hydrolysis in single-strand DNA or positively charged cytosines in an i-motif [14]. The C-rich strand of the hTERT promoter can fold into an i-motif structure (Figure 3), where many of the cytosine bases form hemi-protonated cytosine–cytosine base pairs. The deamination rates of cytosine and its analogs in solution are determined by the fraction of protonated molecules and the hydroxide concentration [24]. Therefore, an increase in the fraction of protonated cytosine bases near physiological pH could exponentially increase the rate of cytosine deamination.

On the other hand, cytosine bases in Watson–Crick base pairs are protected from both protonation and water attack by base pairing and stacking. The p*K_a_* of dCMP is approximately 4.4 [43]; however, in duplex DNA, the p*K_a_* falls to approximately 3.7 as the N3 position of the cytosine base, the site of protonation, is blocked from protonation due to the formation of a hydrogen bond with the guanine imino proton. Prior studies [14,28] have shown that the rate of cytosine deamination in duplex DNA is approximately 100 times slower than in single-stranded DNA.

i-motifs can form in sequences where there are at least four runs of three or more cytosine bases [44]. Phan et al. (2000) have examined the solution structure of a cytidine-rich fragment of the human telomere d[(CCCTA_2_)_3_CCCT] at pH 5 [30]. In the four-stranded i-motif structure, the cytosine base pairs are fully intercalated with one another. The imino proton of the hemi-protonated cytosine base pair is observed at 15 to 16 ppm, downfield from the imino protons of the AT and GC base pairs. Surprisingly, the CH+C imino protons of such structures show slow solvent exchange, indicating they are protected from water [45]. The observed chemical shifts of the non-exchangeable H5 and H6 protons of cytosine bases are influenced by both protonation (downfield shifts) and base stacking (upfield shifts) [24]. In the i-motif structure, the chemical shifts are close to the monomer values [46], indicative of simultaneous base stacking and protonation effects.

The i-motif structure could therefore promote cytosine deamination by protonation but also protect against deamination by shielding or protecting the cytosine C4 position from water attack. We therefore examined the potential accessibility of protonated cytosines by examining a published i-motif structure (Figure 3). We first examined the solvent accessible surface area (SASA) of a cytosine base in an unpaired, single-stranded region with a 1.4 Å water molecule. An individual unpaired cytosine base was found to be accessible to a water molecule with a surface area of 452.4 Å^2^. In contrast, the C4 position of a cytosine in an i-motif structure was found to be completely shielded and not exposed to the solvent (Figure 3B). This result suggests that an i-motif will more likely protect cytosine from deamination as opposed to catalyzing deamination.

### 3.2. Formation of i-Motifs by Regions of the hTERT Promoter as Studied by UV and CD Spectroscopy

The C-rich strand of the *hTERT* promoter has eleven runs of at least three cytosines and therefore could form multiple i-motif structures. We followed the approach of Palumbo et al. (2009), who divided the *hTERT* core promoter G-rich strand into nine fragments, each containing the minimum number of G tracks to form a quadruplex, and assessed their quadruplex-forming ability [10]. We used a similar strategy with the C-rich strand here (Figure 4) to create segments possibly forming an i-motif. Rogers et al. (2018 and 2021) have shown that i-motif formation can be monitored by measuring the pH-dependent UV and CD changes of oligonucleotides in water [47,48].

For this study, oligonucleotides ranging in length from 19 to 27 bases were prepared. These segments (I–IX) correspond to those examined for the complementary G-rich strand by Palumbo et al. (2009) [10]. Each sequence had at least three runs of at least three consecutive C’s, which fulfills the minimum requirement for i-motif formation. Oligonucleotides were dissolved in a buffer and titrated to a defined pH as measured with a pH meter. Both UV spectra and CD spectra were obtained as a function of pH at 20 °C, as shown in Figure 4 and the associated Appendix A). The pH-dependent changes in the CD spectra were then plotted versus pH to obtain a titration curve. The apparent p*K_a_* for each system was then determined by fitting the observed data to the Henderson–Hasselbach equation.

The observed p*K_a_* values appear in Appendix A and range from 5.05 to 6.20. As the pK of 2′dCMP is ~4.4, i-motif formation is indicated by the measurement of an observed pK_a_ > 4.4. The higher the apparent p*K_a_*, the more likely it is that an i-motif can form under physiological conditions. Sequences forming an i-motif with apparent p*K_a_* values of 7 have been observed for other C-rich sequences [48]. The oligonucleotides that are most likely forming an i-motif are sequences I, II and IX, with apparent p*K_a_* values of 6.00, 6.20 and 6.20. These oligonucleotides are on the 5′ and 3′ ends of the C-rich sequence. The deamination hotspot at C250 is within i-motif-forming oligonucleotides I and II, while the hotspot at C228 is not within any of the regions with higher i-motif formation. While the study reported here examines the propensity of fragments of the C-rich promoter sequence to form i-motifs, the full-length sequence could form a multitude of i-motif structures using adjacent or non-adjacent CCC sequences.

In order to observe possible i-motif formation by the CG-rich *hTERT* promoter region, a 123-base-long oligonucleotide containing the 68-base CG-rich region was constructed (Figure 2). The CD spectra of this oligonucleotide were then measured in solution as functions of temperature and pH (Figure 5). While such data could suggest i-motif formation within the C-rich region, it cannot establish which of the possible regions studied above might be involved. At pH 7.0, little change in the CD spectra is observed between 20 °C and 90 °C, suggesting little or no i-motif formation. At pH 6.0, temperature-dependent changes are observed, with a midpoint of 42 °C. At pH 5, a midpoint of 74 °C is observed. These data suggest that the C-rich strand of the hTERT promoter could form an i-motif at physiological temperatures below neutral pH; however, such a structure is unlikely at physiological pH and temperature.

### 3.3. Cytosine Deamination as Measured by DNA Sequencing

To understand the landscape of cytosine deamination for all cytosines of the C-rich promoter sequence, we utilized NGS. We designed an oligonucleotide comprised of the G- and C-rich *hTERT* promoter, flanked by primers containing Illumina Nextera sequencing adapters. We prepared the two complementary oligonucleotides by solid-phase DNA synthesis so that we could examine the C-rich strand in isolation and when paired with the G-rich strand. Prior studies have examined cytosine deamination at one or a few positions. We wished to determine if deep sequencing might allow simultaneous determination of deamination rates at all 46 of the cytosines in the *hTERT* promoter.

Following incubation, oligonucleotides were amplified by PCR and then sequenced with the MiSeq instrument. From the sequence reads, those with low confidence scores were eliminated. The resulting data set, comprising at least 100,000 reads per nucleotide position, was tabulated as indicating the number of A, C, G or T reads at each position. Transition mutations were scored as C-to-T, T to C, A to G and G to A changes relative to the wild-type sequence. Observed sequence changes could represent the background mutation frequencies arising from prior damage to the DNA templates as well as errors that occur during the DNA sequencing process. We therefore incubated the 123-base sequence in a buffer at 4 °C both as a single strand and when duplexed with the G-rich complementary strand to establish the sequence-dependent background transition mutation frequencies. Additional aliquots of the oligonucleotides, both single-strand and duplex, were incubated at selected pH values at 37 °C or 95 °C.

In Figure 6, the frequency of transition mutations observed at each sequence position in the 68-base region is shown along the sequence of the *hTERT* promoter for a single-stranded oligonucleotide heated at pH 7 and 95 °C for 4 h. Under these conditions, most secondary structures would be lost. Interestingly, transition mutations are observed at all cytosine positions, of similar magnitude, and significantly higher than transition mutations at A, T and G positions (*p* < 0.0001). These mutations can be attributed to cytosine deamination to uracil, which codes as T during DNA sequencing.

We measured the average deamination frequency across 46 cytosines in this sequence to be 1.78 × 10^−3^ ± 2.73 × 10^−4^, (15.3%), which corresponds to a rate constant at 95 °C, pH 7.0, of 1.23 × 10^−7^ s^−1^ and was significantly higher than deamination rates obtained at all pHs at 37 °C (*p* < 0.0001) (Appendix A). This value is in agreement with the prior values of Lindahl and Nyberg (1974) of 2 × 10^−7^ s^−1^ and Frederico et al. (1993) of 1.3 × 10^−7^ at 90 °C for single-stranded DNA [14,28]. These data suggest that deep DNA sequencing can be used to measure sequence-specific deamination rates. Our results reveal that, in the absence of a secondary structure, all cytosines in the hTERT promoter region examined are equally susceptible to hydrolytic deamination.

We next measured deamination frequencies for the C-rich strand incubated at 37 °C and pH 7, 6, and 5 for 11 d. At a reduced pH and temperature, the i-motif would likely form and potentially modulate deamination rates. Results are shown in Figure 7. At the top is the frequency of transition mutations measured at pH 7, 37 °C (average frequency 4.94 × 10^−4^ ± 2.13 × 10^−4^, 43.32%) minus background mutation. In contrast to the 95 °C data (Figure 6), deamination frequencies at specific sequences vary to a greater extent from the average. but none were considered outliers by ROUT analysis. The positions at which C-to-T transitions cause activation of the *hTERT* promoter (C250 and C228) are closer to the average and do not appear to deaminate at unusually high rates. Similar impressions are revealed by data obtained from incubation at pH 6 (middle, average frequency 3.65 × 10^−4^ ± 1.57 × 10^−4^, 42.60%) and pH 5 (lower, average frequency 8.31 × 10^−4^ ± 3.38 × 10^−4^, 40.60%). Although cytosine in the center of CCC or CCC tracts might have higher deamination rates, no patterns of statistical significance were identified. At pH 5, 37 °C, we would expect the i-motif to form, and we observed a significant increase in the transition rate compared to pH 6 and 7 (*p* < 0.0001). The deamination frequencies at C250 and C228 are, in all cases, similar to the averages, as are the Cs in the sequence TCCG, which could not be involved in an i-motif. These data suggest that the potential formation of an i-motif does not significantly influence observed deamination rates at any particular site. The average deamination rate constant at pH 7.0 and 37 °C was measured to be 5.2 × 10^−10^ s^−1^. Previously, Ehrlich et al. (1986) measured cytosine deamination rates at higher temperatures [27]. They proposed by extrapolation that the corresponding rate constant at 37 °C would be 2.1 × 10^−10^ s^−1^, in close agreement with the value reported here.

In the final study, we prepared a duplex structure by mixing equimolar amounts of the C- and G-rich strands, followed by brief heating and cooling. Duplexes were then incubated at 37 °C for 11 d, isolated and sequenced. The data are presented in Figure 8 (upper pH 7.0, average frequency 3.81 × 10^−4^ ± 2.62 × 10^−4^, 68.67%; middle pH 6, average frequency 3.83 × 10^−4^ ± 2.50 × 10^−4^, 65%; lower pH 5, average frequency 6.74 × 10^−4^ ± 2.24 × 10^−4^, 33%). As with the data obtained with the single strand, deamination frequencies vary from position to position, but no outliers were identified by ROUT. The average deamination rate of the pH 5 treatment was again significantly higher than for pH 6 or pH 7 (*p* < 0.0001). Observed frequencies for the C250 and C228 positions, as well as the TCCG sequence, are close to the average. Similar data were obtained for duplexes incubated at pH 6 and pH 5. The measured rate constant for deamination in the duplex measured at pH 7, 37 °C, is 4.0 × 10^−10^ s^−1^, lower than the value measured for the single strand but not significantly different. Comparisons of average deamination rates are summarized in Figure 9.

## 4. Conclusions

Transition mutations (C-to-T) within the C-rich strand of the *hTERT* promoter arising from cytosine deamination result in the activation of *hTERT* expression, increased tumor cell survival and reduced patient survival. Understanding conditions that might modulate sequence-dependent deamination rates is therefore important for revealing the etiology of *hTERT* promoter mutations. The CG-rich *hTERT* promoter sequence is unusual in that the G-rich strand can form a quadruplex or series of quadruplex structures at physiological pH, while the C-rich strand could form an i-motif with protonated cytosine bases at and below neutral pH.

In this study, we confirmed that some sequences across the C-rich strand are capable of forming i-motif structures. As cytosine protonation promotes deamination with monomers of cytosine and analogs in solution, we wished to investigate if cytosine protonation and i-motif formation could induce deamination. We therefore developed an approach using deep DNA sequencing to investigate deamination mutations in the C-rich strand. We observed increased C-to-T transition mutations with increasing temperature and time. We did not observe increased mutation frequencies at either of the C250 or C228 sites frequently found to be mutated in human tumors. The observed mutation rates measured with the sequencing method are in agreement with previously reported values from other laboratories. Surprisingly, we did not measure decreased mutation frequencies when the C-rich strand was incubated with the G-rich complementary strand compared to the C-rich strand alone. This result is in accord with the previous report from Lindahl and Nyberg (1974) that cytosine deamination in poly dC-dG was not substantially lower than that measured for poly dC [14]. Structural studies on sequences capable of forming non-duplex structures suggest that multiple configurations are likely to co-exist in equilibrium with one another (Figure 10). The C-rich strand of the *hTERT* promoter is likely single-stranded much of the time under physiological conditions despite the presence of the complementary G-rich strand. This could be potentiated by the formation of multiple stable G-quadruplexes in the G-rich strand, which would prevent duplex B–DNA formation (Figure 10C,E) [10].

Our study reveals that the sequence positions where activating mutations occur are not deamination hotspots and are unlikely related to the oligonucleotide conformation. Instead, activating mutations are selected by their capacity to increase *hTERT* transcription. Our data does suggest, however, that cytosine bases in sequences with the capacity to form complicated structures frequently found in promoter regions across the genome are more single-stranded than duplex and will therefore deaminate up to 100 times faster than those in duplex DNA [14,28].

## Figures and Tables

**Figure 1 biomolecules-13-01308-f001:**
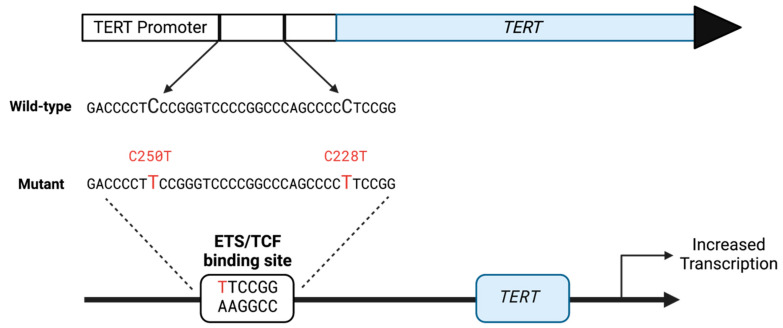
The *hTERT* gene indicates the position of the hotspot C-to-T transition mutations. The deamination of C-to-T generates a high-affinity binding site for the ETS transcription factor that drives transcription of the *TERT* gene.

**Figure 2 biomolecules-13-01308-f002:**
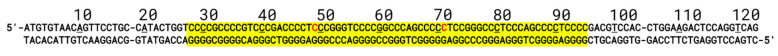
Sequence of the oligonucleotide used in this study for sequencing, which includes Seq-hTERT-C-rich. The *hTERT* core promoter sequence is highlighted, and mutational hotspots are in red. A “-” denotes the boundary between primer annealing sites.

**Figure 3 biomolecules-13-01308-f003:**
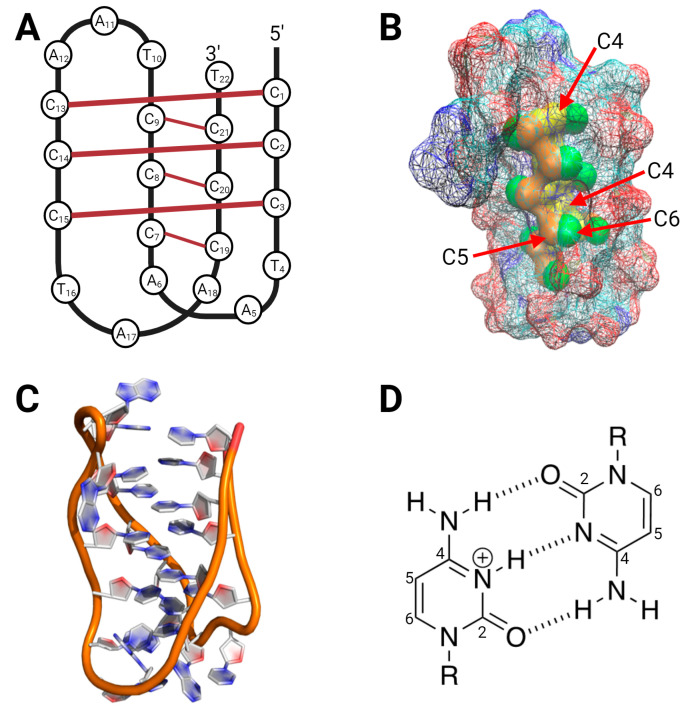
Model of the i-motif used for the determination of solvent-accessible surface area. (**A**) Schematic representation of an i-motif d[(CCCTTA_2_)_3_CCT]. (**B**) i-motif PDB: 1ELN surface area was probed with a 1.4 Å water molecule and was meshed to allow visualization of internal C4-6 atoms. The C4 (yellow) atom is buried in the core of the i-motif and is inaccessible to a water molecule. The cytosine carbon 4 (C4) is in yellow, C5 in orange, and C6 in green. (**C**) i-motif PDB: 1ELN stick figure. (**D**) Representation of a C:C+ base pairing within an i-motif.

**Figure 4 biomolecules-13-01308-f004:**
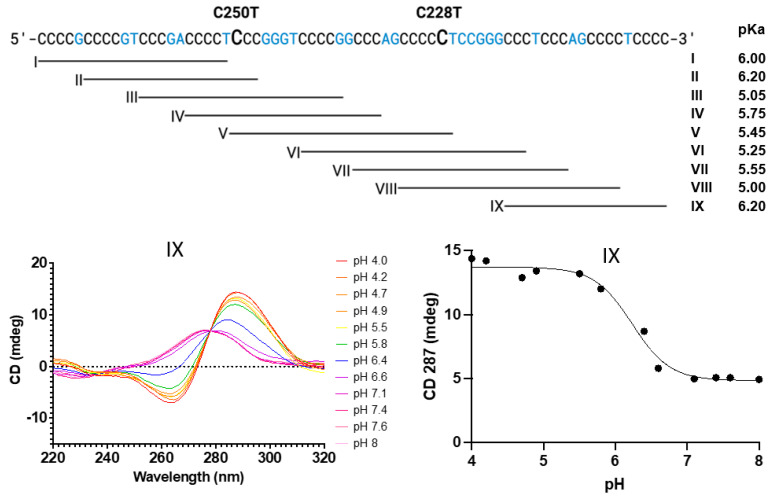
The formation of an i-motif can be monitored by CD = f(pH). At the top is the sequence of the C-rich *hTERT* promoter region and the nine oligonucleotide segments (I to IX) examined for i-motif formation. The CD spectrum of each oligonucleotide was examined in solution as a function of pH. Representative spectra (**left**) of a single oligonucleotide and the corresponding CD titration (**right**) are shown in the lower portion of the figure.

**Figure 5 biomolecules-13-01308-f005:**
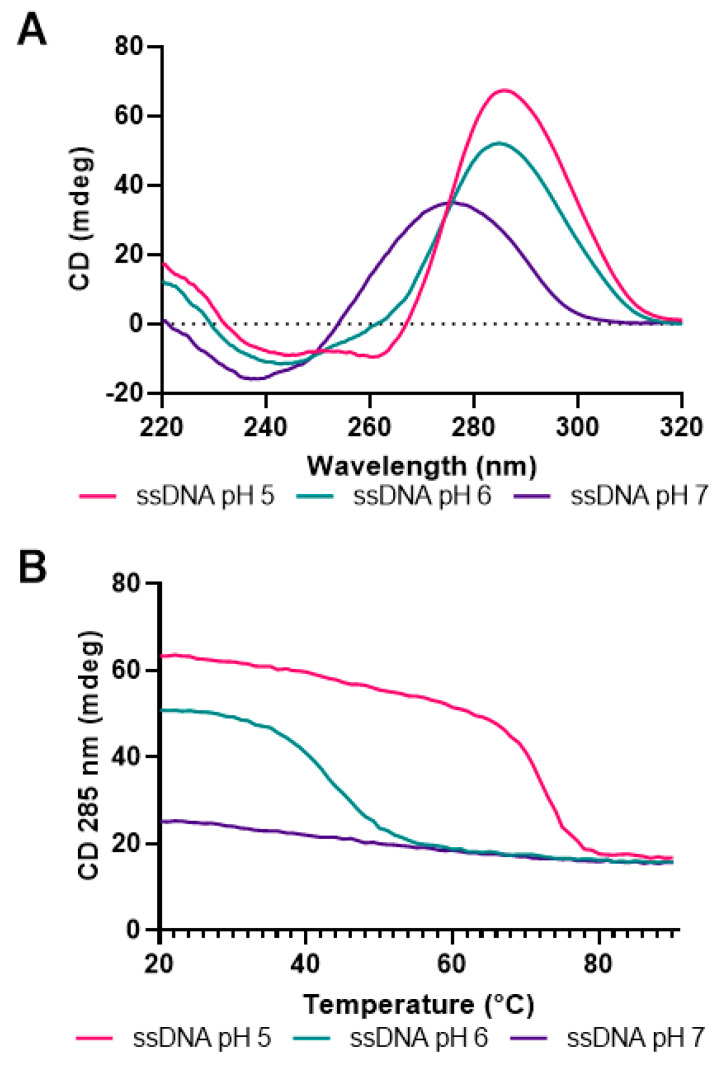
The C-rich *hTERT* promoter can form an i-motif at pH 6 that is stable to 42 °C. At pH 5, the i-motif is stable up to 74 °C. (**A**) CD spectra of the single-stranded 123 nt oligonucleotide used in sequencing reactions at three values of solution pH. (**B**) i-motif melting is indicated by a decrease in CD absorbance at 285 ± 5 nm as a function of temperature. The melting temperature was determined using the first derivative.

**Figure 6 biomolecules-13-01308-f006:**
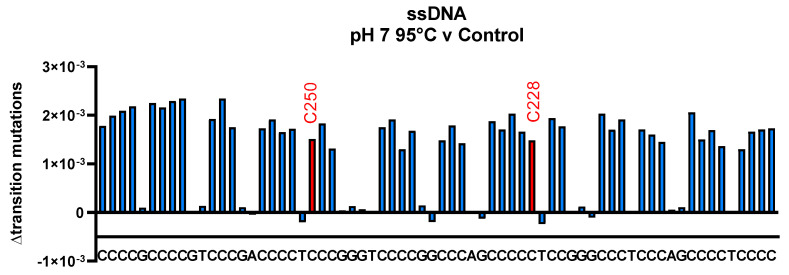
Cytosine deamination frequencies are similar across all 46 cytosines within the 68-base *hTERT* promoter region in a single-stranded oligonucleotide heated at pH 7, 95 °C. The 123-base pair-DNA sequence (Figure 2) was incubated in a buffer at pH 7 at 95 °C for 4 h or maintained at 4 °C. The oligonucleotide was then amplified by PCR and sequenced. The sequence of the 68-base promoter region is shown as the X-axis in the figure above. The fraction of transition mutations at each sequence position was determined by subtracting the transition mutation frequency measured at 4 °C from the mutation frequency measured at 95 °C.

**Figure 7 biomolecules-13-01308-f007:**
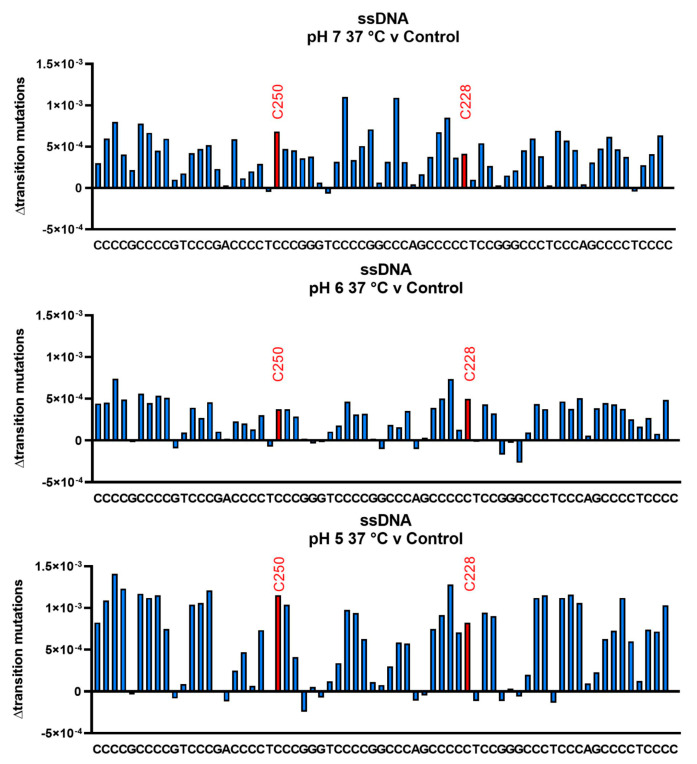
Measurement of deamination frequencies for single-stranded oligonucleotides incubated at 37 °C and pH 7, pH 6 and pH 5 shows greater site variability, but hotspot positions do not deaminate faster than the average of all other cytosines. Single-stranded oligonucleotides were incubated in a buffer at a defined pH of 37 °C for 11 d. Oligonucleotides were sequenced, and deamination frequencies were measured by subtraction of the pH 7, 4 °C control. Averages and standard deviations appear in Appendix A. The average deamination frequency at pH 7, 37 °C and 11 d was measured to be 4.94 × 10^−4^, which corresponds to a deamination rate constant of 5.2 × 10^−10^ s^−1^. Previously, Ehrlich et al. (1986) extrapolated a rate constant for the deamination of cytosine in a single-stranded oligonucleotide at 37 °C to be 2.1 × 10^−10^ s^−1^, which is close to the value reported here [27].

**Figure 8 biomolecules-13-01308-f008:**
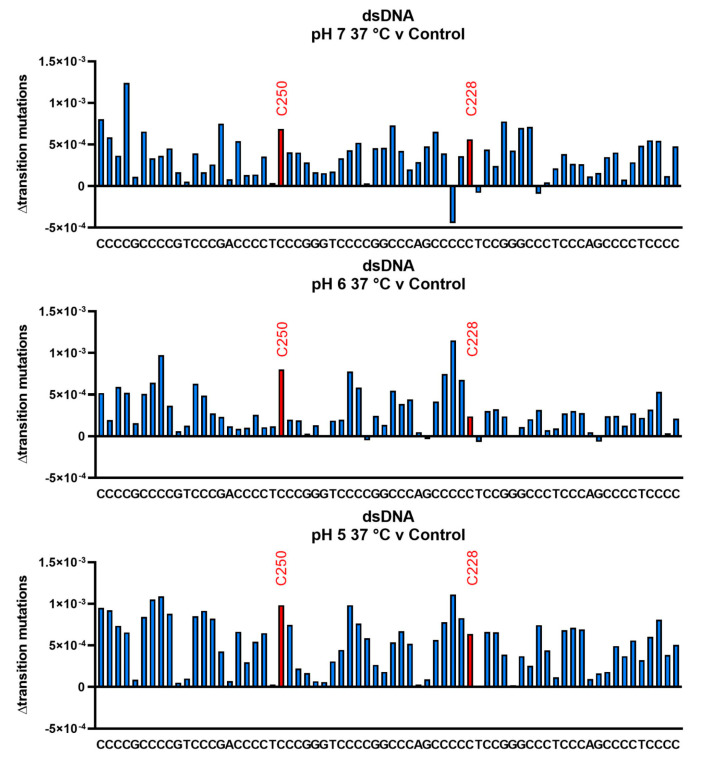
Deamination rates measured in duplex DNA are not slower than in the single strand. The C-rich oligonucleotide from the *hTERT* promoter was incubated, in the presence of its G-rich complement, at 37 °C for 11 d at pH 7, 6 and 5. The oligonucleotides were sequenced, and transition mutation frequencies were obtained by subtracting the results from those obtained with an oligonucleotide duplex incubated at 4 °C. Sequence-dependent variations are observed, but the hotspot positions at C250 and C228 do not deviate significantly. The average transition mutation frequency at pH 7 is 3.81 × 10^−4^, corresponding to a rate constant of 4.0 × 10^−10^, which is not substantially different than the single-stranded rate constant (Appendix A). In contrast, a previous study [29] showed deamination rate constants in duplex DNA to be more than 100-fold lower.

**Figure 9 biomolecules-13-01308-f009:**
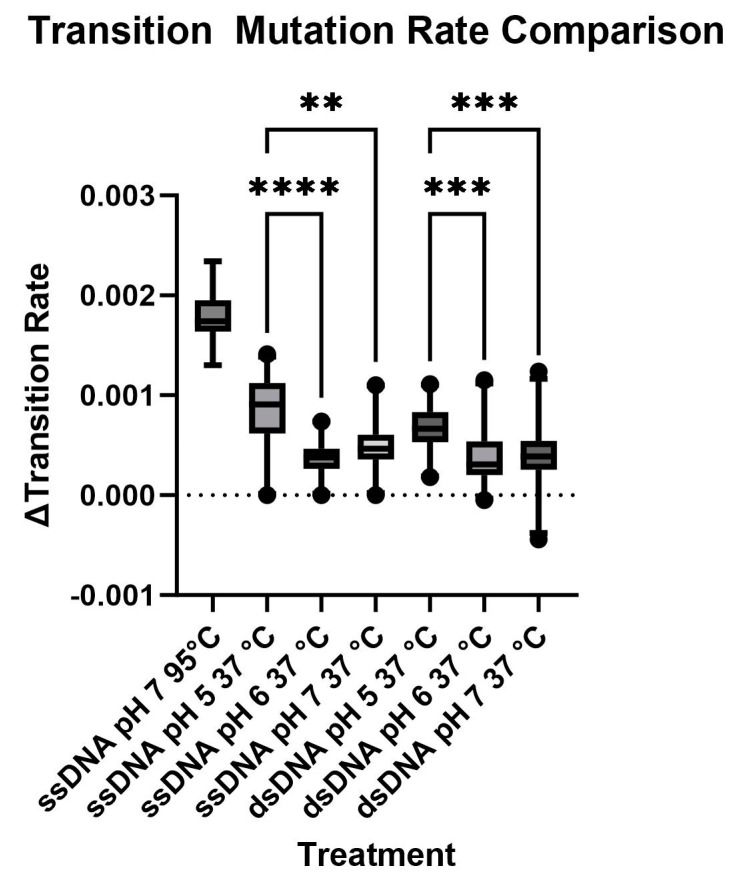
High temperatures and low pH significantly increased the transition rates of the C-rich strand, regardless of the presence of the complementary strand. The Kruskal–Wallis test showed that there are overall significant differences between the treatment conditions (**, *p*-value < 0.01; ***, *p*-value < 0.001; ****, *p*-value < 0.0001). Dunn’s test was used to identify pairwise differences; statistical differences are partially annotated (Appendix A). There are significant differences due to incubation temperature and changes in pH from 5 to 6 or 7. No significant differences were observed between ssDNA and dsDNA at the same incubation conditions. The ssDNA incubated at 95 °C had a transition rate significantly higher than all other treatments, and asterisks are not shown for clarity.

**Figure 10 biomolecules-13-01308-f010:**
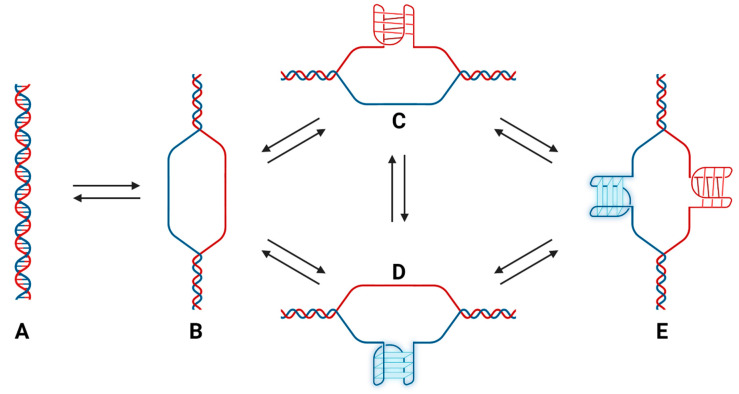
The *hTERT* promoter is likely involved in a complex equilibrium under physiological conditions that involves duplex, quadruplex and i-motif structures that continuously interconvert. Cytosine residues are protected from deamination in duplex (**A**) and i-motif configurations (**C**) due to reduced solvent accessibility, but when interconverting, they are more exposed than in single-stranded DNA (**B**,**D**) [12]. Potentially, G-quadurplex formation (**D**) and i-motif formation (**C**) can transiently occur on both strands during the interconversion process (**E**). The data presented here suggests that i-motif formation does not enhance deamination rates over rates in single-stranded DNA, and in particular, deamination is not enhanced at C250 and C228. However, due to the persistent structural fluctuations, all cytosine bases in the promoter are ~100 times more susceptible to deamination than if in duplex DNA. Cells in which transition mutations at C250T and C228T occur show increased TERT expression and enhanced survival.

## Data Availability

Appendix A is available online. Sequencing data are available at the National Institutes of Health BioProject, accession: PRJNA988536.

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
