# Peer review of "Transition Mutations in the hTERT Promoter Are Unrelated to Potential i-motif Formation in the C-Rich Strand"

_biomolecules, 2023, doi:10.3390/biom13091308_

Round 1

Reviewer 1 Report

Overall, this manuscript focuses on C-to-T transition (deamination) mutations in the hTERT promoter region. By using biophysical study and deep sequencing, the results reveal that the sequence positions where activating mutations occur are unlikely related to the oligonucleotide conformation, which is intriguing and significant to i-motif study. This work can be considered to be published in Biomolecules. However, the following concerns should be addressed before publication.

1. In Figure S1, the pH-dependent curves of III, VII, and VIII sequences look different from the other sequences (not standard S-curve). Can the authors give a reason? Is it related to the C-tracks? Related to the mutation point?

2. In Figure 3B, the authors write “the C4 position of a cytosine 311 in an i-motif structure was found to be completely shielded and not exposed to the solvent”. Please refine the figure by indicating where the C4 position is.

3. In Figure 5b, the melting curves at different pH values need to be fitted and the melting temperatures (Tm) should be calculated. The authors estimate the midpoint which can reflect the melting temperature but not accurate. Please use first derivative to find Tm values.

Minor editing of English language required.

Reviewer 2 Report

In this study, the authors focused on investigating the potential role of DNA secondary structure in the promoter of the hTERT gene, and they hypothesized that the i-motif formation at the C-rich strand of hTERT promoter may promote cytosine deamination and C>T mutations. Using computational modeling, the authors show the cytosine C4 position is shielded in an i-motif structure, suggesting i-motifs are unlikely to promote deamination. They further use deep sequencing to measure deamination rates simultaneously at all 46 cytosines in the promoter while no positions showed substantially increased deamination, including the known hotspots C250T and C228T. Overall, this is a technically robust study combining computational modeling, biophysical techniques, and deep sequencing methods to provide direct evidence that structural effects do not drive the mutational hotspots in the hTERT promoter. However, I have some concerns as listed below.

Major comments:

1.       In the deep sequencing results, the authors measure the deamination rate based on the average frequency across the whole hTERT core promoter sequence and compare the rates between different groups. The authors should specify the statistical test used to analyze the differences in deamination rates between groups. Besides, considering the i-motif structure is sequence-dependent, it would be informative to calculate deamination rates for each cytosine tract at different locations in the promoter.

2.       In lines 505-508, the authors claim the data show ssDNA deaminates ~100 times faster than duplex DNA. However, they do not specify the supporting data source for this comparison. Moreover, in lines 494-495, the authors state they "did not measure decreased mutation frequencies" for ssDNA versus dsDNA, which seems contradictory. The authors should clarify the evidence for how deamination rates differ between ssDNA and dsDNA configurations.

Minor comments:

3.       Line 273-274, any literature to support this statement “Mutations that activate the hTERT promoter most likely arise from the deamination of specific cytosine bases in the C-rich strand”?

Round 2

Reviewer 2 Report

The authors have adequately addressed my concerns. No more comments.